# *In silico* epitope prediction and evolutionary analysis reveals capsid mutation patterns for enterovirus B

**Hui Wang**⊚, **Yulu Fang**⊚, **Yongtao Jia, Jiajie Tang, Changzheng Dong**[ID]*

Zhejiang Provincial Key Laboratory of Pathological and Physiological Technology, School of Public Health, Health Science Center, Ningbo University, Ningbo, 315211, China

⊚ These authors contributed equally to this work.
* dongchangzheng@nbu.edu.cn

**Data Availability Statement:** All relevant data are within the paper and its Supporting Information files.

## Abstract

Enterovirus B (EVB) is a common species of enterovirus, mainly consisting of Echovirus (Echo) and Coxsackievirus B (CVB). The population is generally susceptible to EVB, especially among children. Since the 21st century, EVB has been widely prevalent worldwide, and can cause serious diseases, such as viral meningitis, myocarditis, and neonatal sepsis. By using cryo-electron microscopy, the three-dimensional (3D) structures of EVB and their uncoating receptors (FcRn and CAR) have been determined, laying the foundation for the study of viral pathogenesis and therapeutic antibodies. A limited number of epitopes bound to neutralizing antibodies have also been determined. It is unclear whether additional epitopes are present or whether epitope mutations play a key role in molecular evolutionary history and epidemics, as in influenza and SARS-CoV-2. In the current study, the conformational epitopes of six representative EVB serotypes (E6, E11, E30, CVB1, CVB3 and CVB5) were systematically predicted by bioinformatics-based epitope prediction algorithm. We found that their epitopes were distributed into three clusters, where the VP1 BC loop, C-terminus and VP2 EF loop were the main regions of EVB epitopes. Among them, the VP1 BC loop and VP2 EF loop may be the key epitope regions that determined the use of the uncoating receptors. Further molecular evolution analysis based on the VP1 and genome sequences showed that the VP1 C-terminus and VP2 EF loop, as well as a potential "breathing epitope" VP1 N-terminus, were common mutation hotspot regions, suggesting that the emergence of evolutionary clades was driven by epitope mutations. Finally, footprints showed mutations were located on or near epitopes, while mutations on the receptor binding sites were rare. This suggested that EVB promotes viral epidemics by breaking the immune barrier through epitope mutations, but the mutations avoided the receptor binding sites. The bioinformatics study of EVB epitopes may provide important information for the monitoring and early warning of EVB epidemics and developing therapeutic antibodies.

**Funding:** This work was supported by the Zhejiang Provincial Natural Science and Public-interest Foundation of China, Grant Recipient: Changzheng Dong, Award Number: LGF18C060001, URL: https://zjnsf.kjt.zj.gov.cn/portal/index.html; the Ningbo Public-interest Research Project, Grant Recipient: Changzheng Dong, Award Number: No.2021S135, URL: https://kjj.ningbo.gov.cn/. The funders had no role in study design, data collection and analysis, decision to publish, or preparation of the manuscript.

**Competing interests:** The authors have declared that no competing interests exist.

## Introduction

Human enteroviruses belong to the family *Picornaviridae* and the genus *Enteroviruses*, including enteroviruses A (EVA), enteroviruses B (EVB), enteroviruses C (EVC) and enteroviruses D (EVD). EVB mainly consists of Coxsackievirus B1-B6 (CVB1-CVB6) and Echovirus (E) [1], which can be transmitted through the respiratory and fecal-oral routes [2]. Most people have no or mild symptoms after infection, but sometimes it can cause serious diseases, such as viral meningitis, myocarditis, and neonatal sepsis [2,3]. According to the 2014–2016 surveillance report of the US Centers for Disease Control and Prevention, six of the top ten enteroviruses causing outbreaks belong to EVB (E30, E18, CVB3, E9, E11, and CVB5) [4]. In the Chinese population, the overall seroprevalence of the three types of EVB (E6, E30 and CVB5) ranged from 42.9% to 77.5% [5], indicating that EVB infections were common. Among the enteroviruses isolated from viral meningitis cases in Zhejiang Province from 2002 to 2018, the top three belonged to EVB (E30, E6, and CVB5) [6]. In recent years, severe EVB-related outbreaks have occurred in many countries around the world [7–11], suggesting the need to strengthen monitoring and early warning of EVB. Currently, there is no available vaccine or antiviral therapy.

Like other enteroviruses, EVB has a single-stranded, positive-sense RNA genome of approximately 7.4 kb. EVB consists of 60 asymmetric subunits, with VP1-VP3 on the outer surface and VP4 on the inner surface of the capsid [12]. The surface of the viral capsid has a deep depression called "canyon" that surrounds the fivefold axes, with bulges called "puff" and "knob" on the rims of the canyon [13].

There are two types of enterovirus receptors: uncoating receptor and attachment receptor. After binding to the uncoating receptor, the virus initiates the uncoating process and releases the RNA genome into the host cell. The binding of attachment receptor can only assist this process, but cannot achieve uncoating alone [12]. The uncoating receptors of Echo and CVB are Human Neonatal Fc Receptor (FcRn) and Coxsackie-adenovirus receptor (CAR), respectively, both of which bind to the virus by insertion into the canyon [12,14]. The decay-accelerating factor (CD55) is the attachment receptor of many EVB serotypes, and its binding site is located on the lateral surface of the canyon [12]. Three neutralizing antibodies (nAbs) of EVB have been reported. The nAbs 4B10 and 6C5 bind to the interior and northern rim of the canyon of E30, respectively. They block the specific binding of E30 to FcRn, thus exerting a neutralizing effect [15]. The nAb 5F5 binds to the southern rim of the canyon and effectively inhibits CAR binding to CVB1, which may be an effective therapeutic or potential preventive agent for CVB1 infection [14]. However, it is unclear whether EVB has more epitopes like EVA [16]. The three-dimensional (3D) structures of native mature particles of some EVB serotypes, including E6, E11, E30, CVB1, CVB3, and CVB5, have been determined [12,14,17–19], laying the foundation for studying the viral pathogenesis, identifying epitopes and developing neutralizing antibodies. In the process of molecular evolution, each serotype has formed multiple clades, and each clade has its genetic characteristics or mutations [20–23]. For example, Han et al. [22] divided the VP1 evolutionary tree of CVB3 into eight clades, A-H, each with a corresponding prevalent region.

Our lab has previously developed a bioinformatics-based prediction algorithm for human enterovirus conformational epitopes, which has been successfully applied to the epitope prediction of EVA [16]. In this study, we aimed to systematically predict the epitopes of EVB based on the 3D structures. Further, we analyzed the role of epitope mutations in the molecular evolution and epidemic, to provide references for the monitoring and early warning of EVB outbreaks and the development of therapeutic antibodies.

## Materials and methods

### Sequence and structure analysis

The 3D structures of the native mature particles of six EVB serotypes have been determined, which are E6, E11, E30, CVB1, CVB3 and CVB5. [12,14,17–19]. We downloaded their PDB files (6ILP, 6LA3, 7C9S, 7DPF, 1COV and 7C9Y) and amino acid sequences (AYD61416, QIZ12960, AKE33312, QQM18106, AAB02228 and QJR83057) of viral proteins (VPs) from the RCSB PDB database [24] (https://www.rcsb.org) and NCBI nucleotide database [25] (https://www.ncbi.nlm.nih.gov/nuccore), respectively. BLAST (https://blast.ncbi.nlm.nih.gov/Blast.cgi) was used to calculate the sequence identities. The amino acid sequences of EVB were aligned using the MUSCLE tool of MEGA X [26]. The PDB files and alignment files were imported into ESPript 3.0 [27] (https://espript.ibcp.fr) to annotate the secondary structure information. In addition, PyMOL [28] was used to map the surface structure of the VPs and calculate the root mean square deviation (RMSD). CE-BLAST [29] was used to measure the antigenic similarity of viruses. All software and tools were set to default parameters.

### Conformational epitopes prediction

We developed a bioinformatics-based prediction algorithm for human enterovirus conformational epitopes, which has been successfully applied to the epitope prediction of EVA [16]. In this study, we used it to predict the epitope of EVB. The three key steps of the algorithm were as followed: (1) The epiprep tool was used to generate multiple chains Chain 1/2/3/4. Each chain consisted of VP1-VP3 and Chain 2/3/4 surrounded the central chain Chain 1. The multiple chains were considered as a whole, replacing each subunit for subsequent conformational epitope prediction. (2) Three classical online tools for epitope prediction, Epitopia [30] (http://epitopia.tau.ac.il/index.html), Ellipro [31] (http://tools.iedb.org/ellipro/) and DiscoTope [32] (http://www.cbs.dtu.dk/services/DiscoTope/) were used to predict the epitopes of the multiple chains, respectively. By selecting Cα atoms whose capsid center distance was higher than the average distance, the residues on the relatively exposed side of the viral capsid were screened. (3) Residues simultaneously above the default parameters of the all three algorithms (Epitopia, Ellipro and DiscoTope) were defined as core epitopes. The surrounding epitopes were required to meet two conditions: (1) Be predicted as candidate epitopes by any two algorithms; (2) Be in the same position as core epitopes of other EVB serotypes. The prediction results consisted of core epitopes and surrounding epitopes.

### Reliability of predicted epitopes

The reliability evaluation of the prediction algorithm and results mainly relies on the data of binding sites of antibodies and receptors (experimental epitopes) in the literature. The prediction reliability can be evaluated by comparing the consistency of predicted epitopes and experimental epitopes. In the previous study, by evaluating the prediction results of serotypes PV1, EVA71, CVA16 and EVD68 [16,33,34], we found that the prediction algorithm and prediction results had high reliability. In this study, all currently reported experimental epitopes of EVB were collected from the literature and compared them with the predicted epitopes. For further visual comparison, the 3D structure of the viral capsid in the form of a sphere was projected onto a flat surface (roadmap) by RIVEM software to compare the distribution of the footprints of predicted and experimental epitopes.

### Molecular evolution analysis of EVB

Nextstrain [35] is a powerful and user-friendly open-source platform using viral genome data to reconstruct the evolutionary and transmission routes. The VP1 and genome sequences of

EVB were downloaded from the VIPR database [36] (https://www.viprbrc.org) and NCBI nucleotide database [25] (https://www.ncbi.nlm.nih.gov/nuccore). The sequences were removed for lacking isolated information, such as collection time and country. The numbers of sequences for each serotype were shown in S1 Table.

Nextstrain embeds with many evolutionary analysis tools: MAFFT [37] was used for multiple sequence alignment; IQ-TREE [38] was used for phylogenetic trees (maximum likelihood method); TreeTime [39] was used to construct and optimize the time-scale phylogenetic trees; Entropy was used to calculate genetic diversity; Clade command annotated the evolutionary clades and mutations. Since the epitopes were distributed on VP1-VP3, only clade mutations on VP1-VP3 were labeled in the figures. Further, these mutations were mapped onto the footprints to analyze the interactions among mutations, predicted epitopes and binding sites of nAbs and receptors.

## Results

### Sequence and structural characteristics of EVB VPs

The amino acid sequence identities of EVB VPs (VP1-VP3) ranged from 70.92% to 82.52% (S2 Table and S1 Fig), with 70.92% ~ 76.21% for Echo and 80.23% ~ 82.52% for CVB. The sequence identities between Echo and CVB were 72.99%-76.89%, indicating that the identities of CVB were higher than that of Echo, and the identities between Echo and CVB were similar to that within Echo (S2 Fig). The length and position of the β-strands of EVB were highly consistent, and the lengths of the loops were also highly similar. Compared to Echo, CVB had 1–3 fewer residues in the VP1 BC loop, 3–8 fewer residues in the VP1 C-terminus, and 2 more residues in the VP2 EF loop (S1 Fig). The structural differences (RMSD) of EVB VPs ranged from 0.339Å to 0.873Å (S2 Table), and the RMSD of Echo and CVB ranged from 0.425Å to 0.724Å and 0.347Å to 0.873Å, respectively. The RMSD of Echo and CVB ranged from 0.339Å to 0.826Å. The RMSD indicated that EVB had a highly similar 3D structure and no significant difference in the 3D structure of Echo and CVB (S2 Fig). Based on the comparison of sequence and 3D structure, the subtle differences in amino acid sequences of Echo and CVB were not reflected in the 3D structure due to the high similarity of the secondary structure.

### Prediction results of EVB conformational epitopes

Similar to EVA [16], the conformational epitopes of EVB were distributed in three sites on the viral capsid surface (Fig 1): site 1, site 2 and site 3, indicating that the three-site distribution was the basic pattern of enterovirus. Site 1 (red) was located in the "north rim" region on the north side of the canyon near the fivefold axis. Site 2 (green) was located in the "puff" region on the south side of the canyon near the twofold axis. Site 3 (blue) was divided into two parts: the "knob" region on the south side of the canyon and the threefold axis area. The prediction results of the EVB epitopes were shown in Table 1 and S1 Fig. A total of 71 residues were predicted as epitopes in eleven secondary structure regions of VP1 (BC loop, DE loop, EF loop, HI loop and C-terminus), VP2 (EF loop and HI loop) and VP3 (knob region of N-terminus, BC loop, HI loop and C-terminus). These regions were also main epitope regions of EVA. In particular, the VP1 BC loop, VP1 C-terminus and VP2 EF loop were the most important epitope regions for all enteroviruses. Compared to EVA and EVD (S3 Fig), EVB had 6–14 more residues in the VP2 EF loop. Compared to EVA, the VP1 DE loop of EVB had 7 more residues, which was close to the length of EVD. In contrast, the VP1 DE loop was a predicted epitope region for EVB and EVD, but not for EVA. Table 1 showed that the distribution pattern of EVB epitopes was highly consistent, with no significant difference between Echo and CVB.

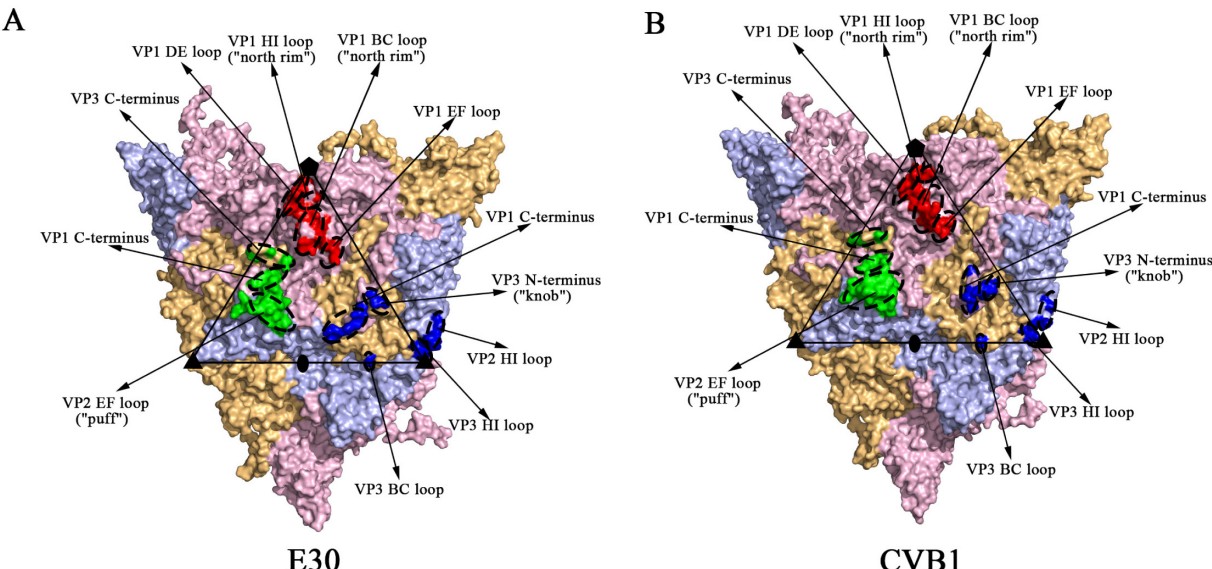

**Fig 1. Multiple chains and the distribution of conformational epitopes of EVB.** Multiple chains and the conformational epitope distributions of E30 (A) and CVB1 (B). VP1-VP3 are indicated in light pink, light blue and light orange, respectively. The conformational epitopes are clustered into three sites, which are represented by red (site 1), green (site 2) and blue (site 3), respectively. Pentagons, triangles and ellipses represent the fivefold, threefold and twofold axes, respectively. The important secondary structures are marked in the figure.

### Predictive reliability of epitopes

We collected all currently reported binding sites of EVB nAbs from the literature. There were three EVB nAbs: nAbs 6C5 and 4B10 (binding to E30) [15], and nAb 5F5 (binding to CVB1) [14]. Fig 2 showed the binding sites of the nAbs to the viral capsid through footprints, indicating that the binding sites of nAbs and the predicted epitopes were highly overlapped. The nAb 6C5 (yellow circles) bound to 11 residues at site 1 of E30, including the VP1 BC loop, DE loop, EF loop, and HI loop, among which the VP1 BC loop was the main binding site of 6C5 (Fig 2A). The nAb 4B10 (blue circles) bound to 8 residues at site 2 of E30, five of which were in the VP2 EF loop, with additional residues in the VP1 C-terminus, VP3 N-terminus and C-

**Table 1. Predicted results of EVB conformational epitopes.**

| VPs | Secondary structure | E6 | E11 | E30 | CVB1 | CVB3 | CVB5 |
|---|---|---|---|---|---|---|---|
| VP1 | BC loop | +++ | +++ | +++ | +++ | +++ | +++ |
|  | DE loop | ++ | +++ | +++ | +++ | +++ | +++ |
|  | EF loop | + | + | + | + | + | + |
|  | HI loop | + | + | + | + | + | + |
|  | C-terminus | +++ | +++ | +++ | +++ | +++ | +++ |
| VP2 | EF loop | +++ | +++ | +++ | +++ | +++ | +++ |
|  | HI loop | ++ | + | ++ | ++ | + | + |
| VP3 | knob of N-terminus | + | + | + | + | + | + |
|  | BC loop | + |  | + | + |  | + |
|  | HI loop | + | + | + | + | + | + |
|  | C-terminus | + | + | + | + | + | + |

+, ++ and +++ indicate that the epitope consists of 1–3, 4–6 and ≥7 amino acid residues, respectively.

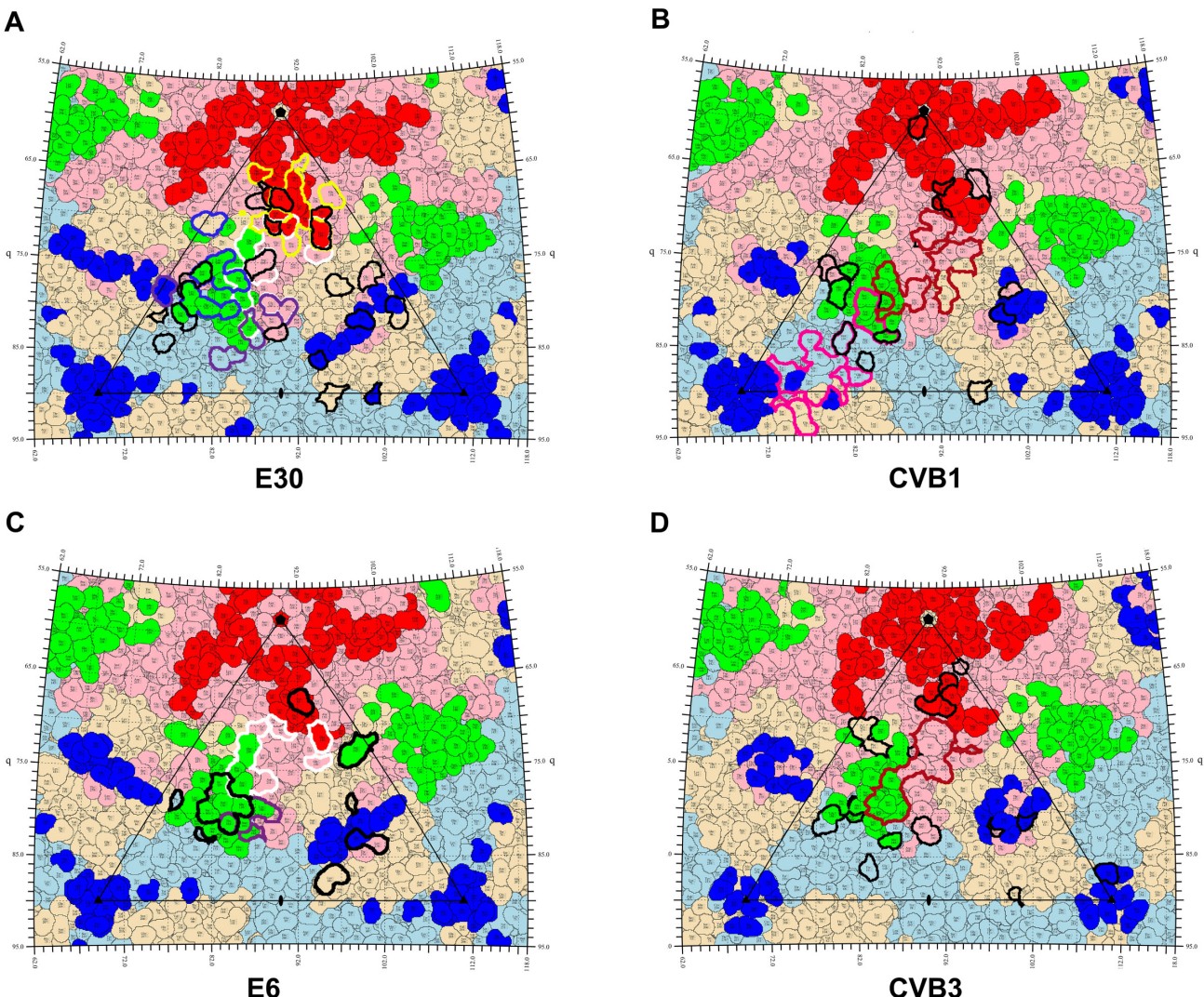

**Fig 2. Footprints of EVB.** VP1-VP3 and epitopes are colored as in Fig 1. The binding sites of nAbs 6C5 and 4B10 to E30 are circled in yellow and blue, respectively (A). The binding sites of nAb 5F5 to CVB1 are circled in pink (B). White circles indicate the binding sites of the uncoating receptor FcRn to E30 (A) and E6 (C), purple circles indicate the binding sites of the attachment receptor CD55 to E30 (A) and E6 (C), and rufous circles represent the binding sites of the uncoating receptor CAR to CVB1 (B) and CVB3 (D). Black circles represent mutation sites (A-D).

terminus (Fig 2A). Wang et al. [15] found that the VP1 GH loop of E30 was difficult to screen for the nAbs that bound to it, which was consistent with our prediction results (the VP1 GH loop was not an epitope region). The binding sites of nAb 5F5 (pink circles) to CVB1 were located between site 2 and site 3, including the VP2 BC loop, EF loop, HI loop and VP3 BC loop, HI loop (Fig 2B). The binding sites highly overlapped with several epitopes, especially the VP2 EF loop.

The uncoating receptor FcRn (white circles) inserted into the canyons of E30 and E6 [12,18], mainly binding to the VP1 BC loop (on the north side of the canyon) and VP2 EF loop (on the south side of the canyon), with additional binding sites in the VP1 EF loop and GH loop (Fig 2A and 2C). The nAbs 6C5 and 4B10 mainly bound to the VP1 BC loop and VP2 EF loop of E30, respectively, competitively blocking the binding of FcRn and triggering the uncoating of Echo. The attachment receptor CD55 bound to the outer surface of the canyon of

E6 and E30 [12,18], mainly to the VP1 GH loop and VP2 EF loop (Fig 2A and 2C). Both nAbs 4B10 and 6C5 had the ability to block CD55 binding to the virus. The nAbs 4B10 and CD55 both bound to the VP2 EF loop of E30. Though the binding sites did not directly overlap, binding to the virus was still spatially blocked by the large body size of nAb 6C5.

The uncoating receptor CAR (rufous circles) also inserted into the canyons of CVB1 and CVB3 and more deeply than FcRn [14,40], mainly binding to the VP2 EF loop and VP1 GH loop on the south side of the canyon, as well as the VP1 BC loop and EF loop on the north side of the canyon (Fig 2B and 2D). The binding of nAb 5F5 could destabilize and disrupt the CVB1 capsid, but the binding sites to CVB1 on the VP2 EF loop did not overlap with CAR. On the one hand, it competed with CAR to bind the virus through steric hindrance. On the other hand, it simulated the uncoating of CAR by triggering conformational changes of the VP2 EF loop to initiate virus destabilization.

Further, the antigenic similarity of EVB was measured using CE-BLAST [29] based on two epitope regions, the VP1 BC loop and VP2 EF loop. It can be found that two epitope regions clearly distinguish CVB and Echo, two species of EVB using different uncoating receptors (S4 Fig), suggesting that they may determine the use of uncoating receptor.

In summary, the binding sites of nAbs and receptors to the EVB capsid overlapped highly with the predicted epitopes, with the VP1 BC loop and VP2 EF loop being the key regions for nAbs and receptors binding.

## Molecular evolution analysis of EVB

Phylogenetic trees based on genome sequences (referred to as genome phylogenetic trees) and VP1 sequences (referred to as VP1 phylogenetic trees) were constructed, respectively. Overall, the genome phylogenetic trees and VP1 phylogenetic trees had similar topologies and evolutionary clades (Figs 3 and 4 and S5–S8), which were consistent with the results previously reported in the literature [20–23,41,42]. However, due to the fewer sequences, genome phylogenetic trees of some serotypes lacked some evolutionary clades. For example, CVB1 lacked clade B (S5B Fig), CVB3 lacked clades C and F (S6B Fig), E6 lacked clades B and D (S7B Fig), and E30 lacked clade A (Fig 4B).

Fig 3A and 3B showed the VP1 phylogenetic tree and genome phylogenetic tree of CVB5, respectively. The genome phylogenetic tree of CVB5 was divided into four major clades, A-D. Except clade D, clades A, B and C shared a common ancestor (Fig 3B). The clade D had five signature mutations: G19S, S95N, V156I, M180I and K200R. Taking V156I as an example, the residuals at position 156 were almost I on clade D, but almost V on other clades (S9A Fig). The clade A-C had four signature mutations: V7I, S273G, E276D, and A279T. Taking E276D as an example, the residuals at position 76 were almost D in clade A-C, while almost E in clade D (S9B Fig). Similarly, A45S, F75Y, N85D and F89I were signature mutations of clade B, and I7V, A90G, V99A, E272K and G273S were signature mutations of clade C. The VP1 phylogenetic tree of CVB5 was also divided into four major clades, A-D. Clades A and B were more proximal on the VP1 phylogenetic tree, while clades B and C were more proximal on the genome phylogenetic tree. This led to some subtle changes in the signature mutations (Fig 3A), mainly in the form of possible changes in the clade where the mutations were located. For example, mutation T279A in clade D of the VP1 phylogenetic tree became mutation A279T in clade A-C of the genome phylogenetic tree. The residues at position 7 of VP1 were I in clades A and B, but V in clades C and D. Due to the changes in the affinity of clades A, B and C, the mutation V7I in clade A-B on the VP1 phylogenetic tree became two mutations, V7I in clade A-C and I7V in clade C on the genome phylogenetic tree. Similarly, the mutation S273G in clade A-B of the VP1 phylogenetic tree became two mutations, S273G in clade A-C and G273S

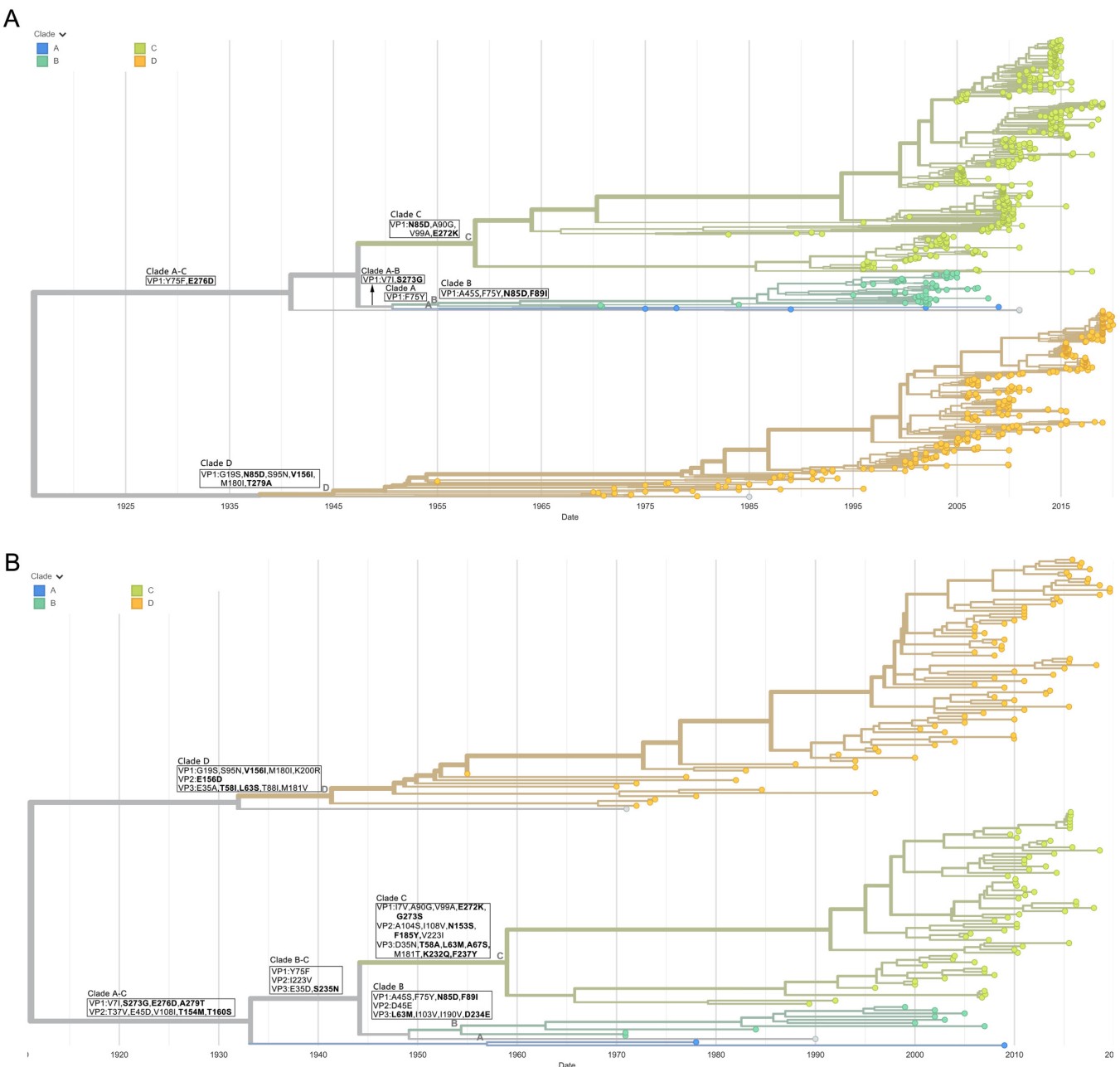

**Fig 3. Phylogenetic trees of CVB5.** CVB5 time-scale phylogenetic trees based on VP1 sequences (A) and genome sequences (B). Mutations are labeled in black frames, with the residues on the epitope regions in bold.

in clade C on the genome phylogenetic tree. Despite these differences, Tables 2 and S3 revealed that the VP1 phylogenetic tree and the genome phylogenetic tree of CVB5 had completely identical mutation positions. This indicated that the signature mutations on the genome phylogenetic tree were reliable despite the relatively fewer sequences. From the genome phylogenetic tree of CVB5 (Fig 3B), it can be found that the emergence of each clade was driven by mutations in the epitope regions. For example, the clade A-C had three mutations (S273G, E276D and A279T) in the VP1 C-terminus and two mutations (T154M and T160S) in the VP2

EF loop. In addition to mutation E156D in VP2 EF loop, clade D had mutations in the VP1 EF loop (V156I) and VP3 knob region (T58I and L63S). The clade B had multiple mutations at epitope regions such as VP1 BC loop, VP3 knob region and C-terminus. The clade B had numerous mutations at the epitope regions such as VP1 C-terminus, VP2 EF loop, VP3 knob region and C-terminus. This showed that the VP1 C-terminus, VP2 EF loop, VP3 knob region and C-terminus were the mutation hotspot regions of CVB5. In addition, several mutations (V7I, G19S and A45S) occurred in the VP1 N-terminus.

Fig 4A and 4B showed the VP1 phylogenetic tree and genome phylogenetic tree of E30, respectively. The VP1 phylogenetic tree of E30 contained eight major clades, A-H (Fig 4A).

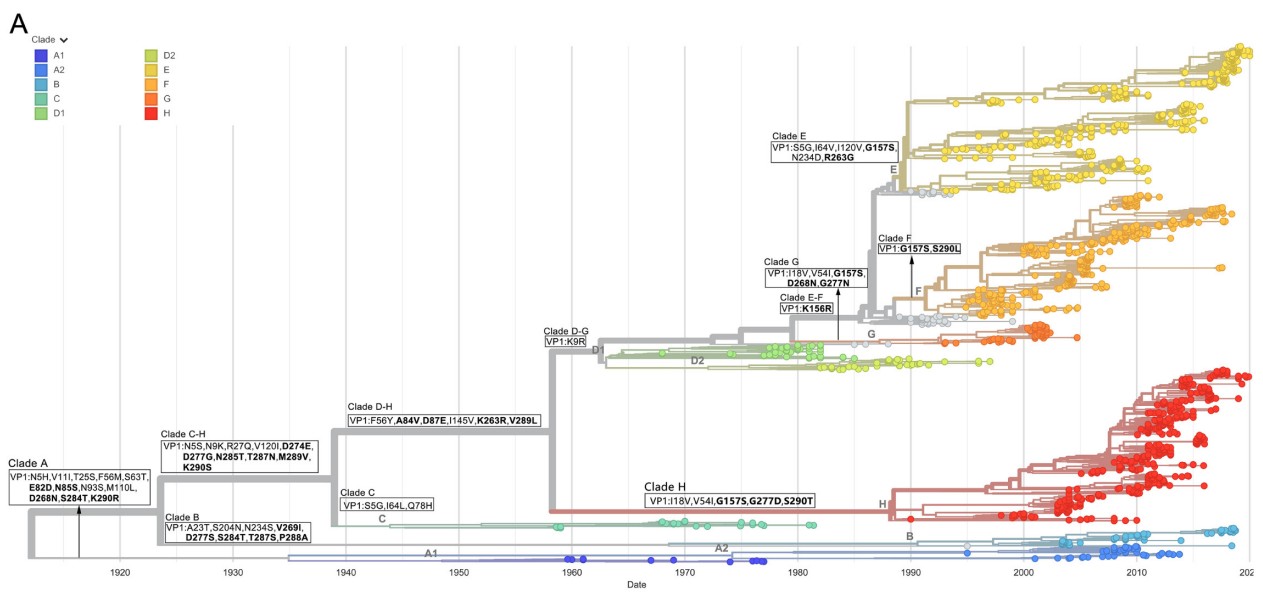

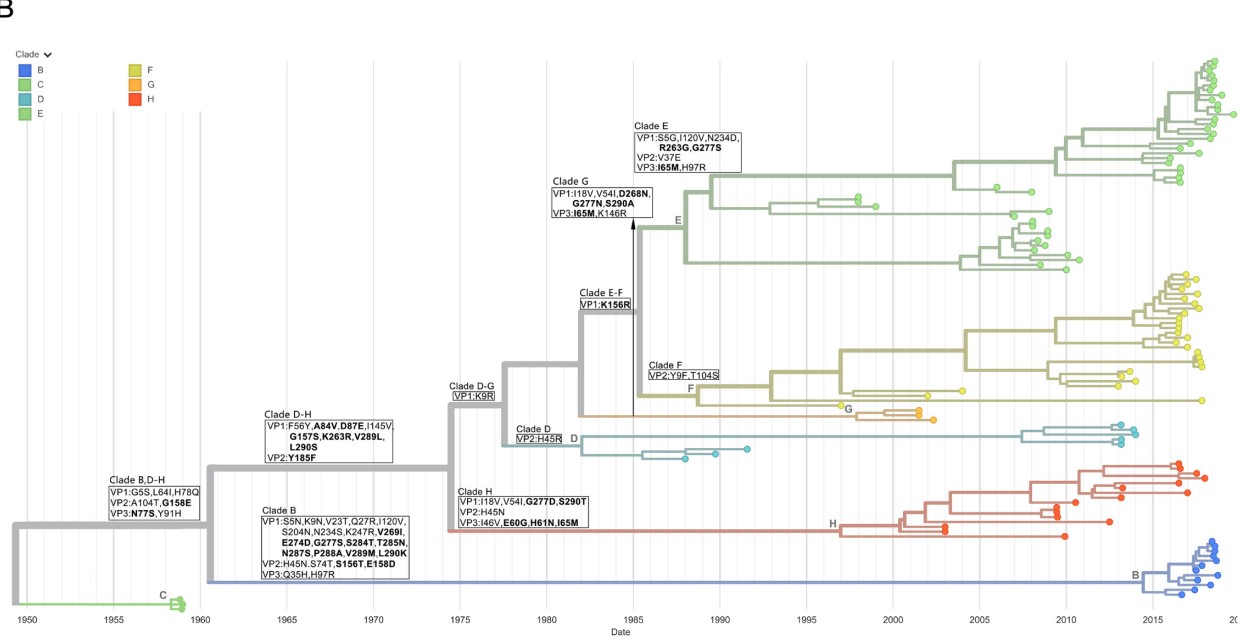

**Fig 4. Phylogenetic trees of E30.** E30 time-scale phylogenetic trees based on VP1 sequences (A) and genome sequences (B).

The topology of the genome phylogenetic tree (Fig 4B) was basically the same as that of the VP1 phylogenetic tree (Fig 4A), but with clade A missing, and clade C was closer to the root of the genome phylogenetic tree. Due to the absence of clade A, most of the signature mutations located in the VP1 N-terminus, BC loop and C-terminus in clade A were also accordingly missing from the genome phylogenetic tree. Three mutations (G5S, L64I and H78Q) in clade B, D-H of the genome phylogenetic tree were located in clade C of the VP1 phylogenetic tree, reflecting the difference between clade C and the other clades. Mutations in clade B of the genome phylogenetic tree, part of which (V23T, S204N, N234S, V269I, S284T, P288A) were located in clade B, and part of which (S5N, K9N, Q27R, I120V, E274D, T285N, V289M, L290K) were located in clade C-H of the VP1 phylogenetic tree. Only ancestral amino acids mutated. Both G277S and N287S became two mutations in clade B and C-H, respectively. Another mutation (K247R) was actually located inside clade B of the VP1 phylogenetic tree. The signature mutations in clades D, E, F, G and H were almost identical in VP1 and genome phylogenetic trees. Comparing Table 2 with S3 Table, it can be found that mutations of the genome phylogenetic tree and the VP1 phylogenetic tree were mostly in the same position. The VP1 N-terminus, C-terminus, VP2 EF loop and VP3 knob region were mutation hotspot regions of E30.

Similar to CVB5 and E30, CVB1, CVB3, E6 and E11 had a large number of mutations on the epitope regions and VP1 N-terminus on almost every clade (Tables 2 and S3 and S5–S8 Figs). The VP1 N-terminus, BC loop, C-terminus and VP2 EF loop were mutation hotspot regions of CVB1 (Tables 2 and S3 and S5 Figs). The VP1 N-terminus, BC loop, VP2 EF loop, and VP3 C-terminus were mutation hotspot regions of CVB3 (Tables 2 and S3 and S6 Figs). The VP1 N-terminus, C-terminus and VP2 EF loop were the mutation hotspot regions of E6 (Tables 2 and S3 and S7 Figs). The VP1 N-terminus, BC loop, C-terminus, VP2 EF loop and VP3 knob region were the mutation hotspot regions of E11 (Tables 2 and S3 and S8 Figs). Further statistics on S3 Table revealed that approximately two-thirds of the mutations were located in the "VP1 N-terminus + epitope" region, half of the mutations were located in the "epitope" region, and one-third to half of the mutations were located in the "VP1 N-terminus + C-terminus + VP2 EF loop" region (S4 Table). For EVB, VP1 N-terminus, C-terminus, and VP2 EF loop were common mutation hotspot regions, in addition to VP1 BC loop, VP3 knob region and C-terminus. The genetic diversity analysis showed the same results that diversities of these mutation hotspot regions were the most (Fig 5).

The EVB mutations (Tables 2 and S3) were further mapped on footprints (Fig 2). Mutations on the viral capsid surface were concentrated in the regions where the three sites of the predicted epitopes were located. Notably, mutations in the receptor binding sites were rare. There were no mutations in the binding sites of the uncoating receptor CAR to CVB1 and CVB3, and the uncoating receptor FcRn to E6. Only the binding sites of FcRn to E30 had three mutations, two of which were located in the north rim (VP1 D87E and K156R) and one next to the puff region (VP1 K263R). There was no mutation in the binding sites of the attachment receptor CD55 to E30, and only one mutation (VP2 T163S) in the binding sites to E6. Although mutations in the receptor binding sites were rare, there were numerous mutations in the surrounding regions. On the one hand, it suggested that EVB broke the immune barrier through epitope mutations; on the other hand, mutations avoided the key region of the receptor binding sites. The effect of the three mutations in the binding sites of FcRn to E30 deserves to be studied in depth. From the E30 phylogenetic tree (Fig 4), all clades with these mutations have become the dominant clades and been epidemic so far, especially the clades with VP1 87E and 156R are absolutely dominant.

**Table 2. Summary of clade mutations of EVB VP1 phylogenetic trees.**

| VPs | Secondary structure | E6 | E11 | E30 | CVB1 | CVB3 | CVB5 |
|---|---|---|---|---|---|---|---|
| VP1 | N-terminus | N5S*,N5S*,V7I, E8D, A10S, M11I, M11V, M18L,K54R, K54R,F56Y, S63A | V7I,N9G,S19G, S23T,T43V, S45G,I48M, K54H, S63T, S69C | N5S,N5H,S5G, S5G,N9K,K9R, V11I,I18V,I18V, A23T,T25S, R27Q,V54I, V54I, F56Y, F56M,S63T, I64L,I64V | M11V, M11V, V12A, V12A, V18I,R21K, I64V | I7V,I7V, I7V, A9S,V18I, G45S,I64V, L68V | V7I,G19S, A45S |
| | BC Loop | R82K,D84A, A84T,D84N, N84D | Y80F,Y80F, Y80H,Y80H, K86E,K86T, T86A,K86Q, T87S,L89R | E82D,A84V, N85S,D87E | T83S,T83S, Y87F | S82T,S84A, K85N,K85T | N85D, N85D, N85D,F89I |
| | DE Loop | A139V, A139T, T141I | | T130S | | I136V | |
| | EF Loop | | V157T,T161A | K156R,G157S, G157S,G157S, G157S | V150I, T155K | | V156I |
| | HI Loop | | P227S,P227S | | | T223A, T223A, T223V | |
| | C-terminus | F266Y,Q277S, I278L,L278I, S281T,S281T, S283T,M286V, T288N,Y289H, Y289H,Y289H | S267T,P268S, N270D,D273E, N276Q,D283E, D283E,T279N, N291T,T291N, Y292H | K263R,R263G, D268N,D268N, V269I,D274E, D277G,D277S, G277D,G277N, S284T,S284T, N285T,T287N, T287P,P288A, M289V,V289L, K290S,K290R, S290T,S290L | N264S, N264S, T266I, S271T, I273S | T277A | E272K, S273G, E276D, T279A |
| | Others | N176S,M181I | S92N,S96N, S96N,M109L, M109I,V117M,M117I, M117I, I186M,I219L, V221M,M238V,V245I, A247V | Q78H,N93S, M110L,V120I, I120V,I145V, S204N,N234S, N234D | I92V,I92V, E106D, E106D | I92L,I92V, I92V,T93S, T94P,A98V, A98V, I110V, I110M, S200A, N202S | Y75F,F75Y, F75Y,A90G, S95N,V99A, M180I |

*The same mutations indicate parallel mutations or reverse mutations in different clades.

## Discussion

Previously, our lab developed a bioinformatic-based prediction algorithm for enterovirus epitopes, successfully predicting the conformational epitopes of EVA and systematically summarizing its distribution pattern [16]. Further, we extended this prediction algorithm to picornavirus, and discovered the key residues of human parechoviruses (HPeV) binding to its nAbs, which is of great significance to understand the pathogenic mechanism [34]. In this study, we focused on EVB. Similar to EVA [16], the predicted conformational epitopes of EVB were all distributed in three clusters (site 1, site 2 and site 3) on the surface of the viral capsid, with the main epitope regions including the VP1 BC loop, C-terminus and VP2 EF loop. The VP1 GH loop was the classic epitope region for EVA, but not for EVB. The EVB uncoating receptors FcRn and CAR [12,14,18,40] were both inserted inside the canyon, so their binding sites to the virus were located at the northern and southern rims of the canyon, including the

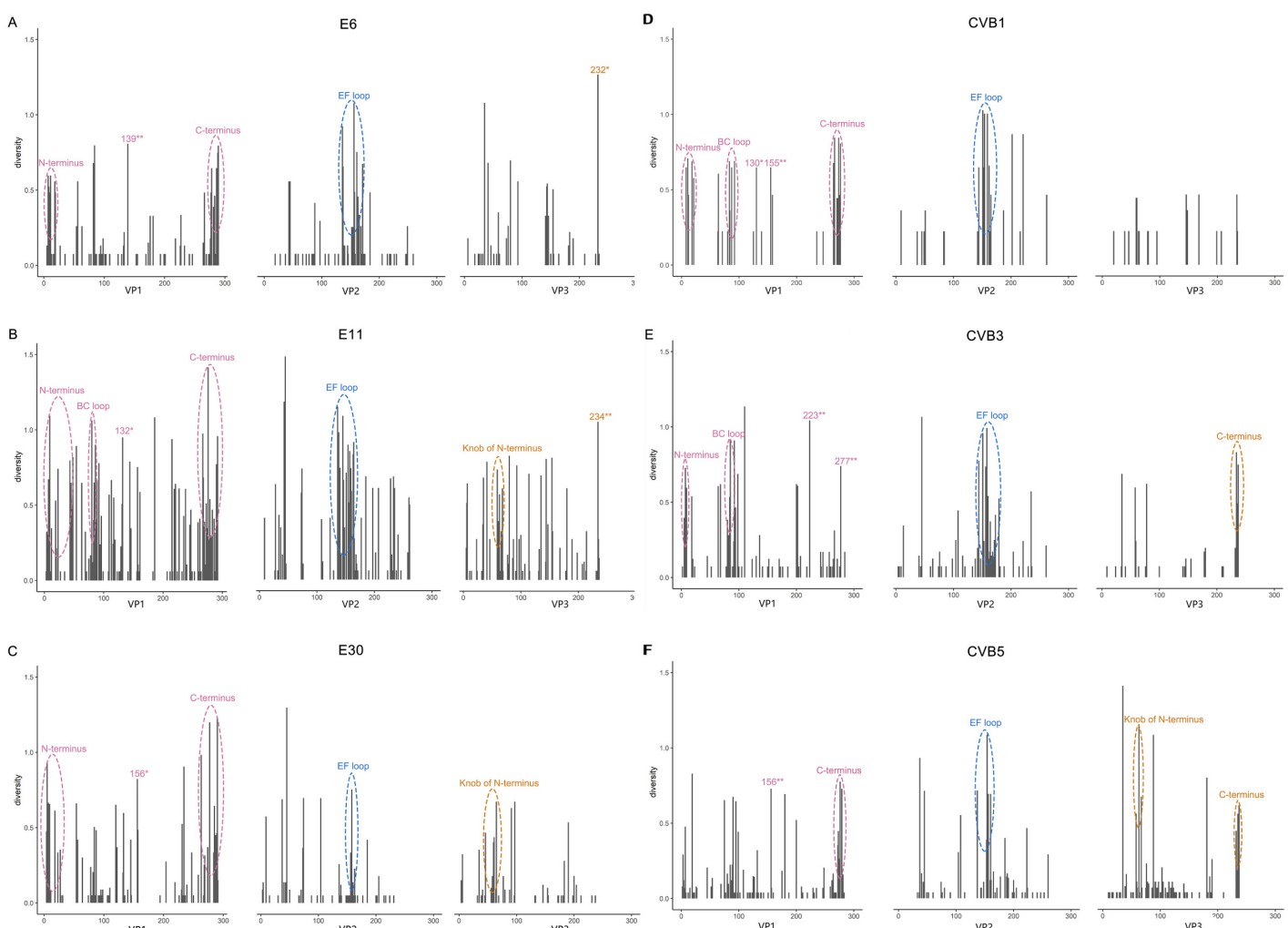

**Fig 5. Genetic diversity of EVB viral proteins.** Hotspot regions of VP1-VP3 genetic diversity are highlighted by pink, blue and brown ovals, respectively. * indicates that the mutation is located in the epitope regions.

epitope regions VP1 BC loop, VP2 EF loop, VP1 EF loop, and several residues of the VP1 GH loop at the bottom of the canyon. The nAbs 6C5 and 4B10 [15] bound to the VP1 BC loop on the north rim and the VP2 EF loop on the south rim of the E30 canyon, respectively, both competitively blocking the binding of the FcRn and initiating the uncoating process. Both 5F5 and CAR [14] were bound to the VP2 EF loop of CVB1. Though their binding sites did not overlap, the block on 5F5 still spatially initiates the destabilization of CVB1 because of the proximity of the two binding sites. EV71 (nAbs 28–7, A9, E19) [43–45], CVA16 (nAbs 18A7, NA9D7) [46], CVA10 (nAb 2G8) [47], CVA6 (nAb 1D5) [48], EVD68 (nAbs 11G1, EV68-228) [49,50] all had nAbs that binding to the VP1 BC loop or VP2 EF loop, indicating that they were common core epitopes of enteroviruses. The VP1 C-terminus was divided into two parts on the viral capsid surface: one immediately adjacent to the puff region and the other near the knob region, both located at the rim of the canyon. No nAb has been reported to bind to these regions in EVB, but has been confirmed as a susceptible region for nAb by other enteroviruses. Wang et al. [51] also found that CVB3 could play an important role in viral pathogenesis by influencing the host cell cycle by nuclear localization through its VP1 C-terminus.

CVB and Echo infected host cells with the help of two different types of uncoating receptors (CAR and FcRn), respectively. From the perspective of VPs identities, the identities of CVB were slightly higher than that of Echo. The sequence variations among different Echo serotypes were large so that the sequence identities within Echo were not higher than that between CVB and Echo; In other words, CVB can be roughly regarded as a "serotype" of Echo. More surprisingly, there was no significant difference in the 3D structures of the structural proteins between CVB and Echo. The VP1 BC loop and VP2 EF loop were critical for binding EVB to the uncoating receptors and neutralizing antibodies. Interestingly, using CE-BLAST [29] to measure the antigenic similarity of EVB on these two epitope regions, it was found that CVB (using CAR as the receptor) was very clearly distinguished from Echo (using FcRn as the receptor) (S4 Fig). It is suggested that CVB and Echo have different antigenic microenvironments on these two epitope regions, and this may determine the use of uncoating receptor.

In order to analyze the role of epitopes in the molecular evolution of EVB, phylogenetic trees were constructed based on the VP1 sequences and genome sequences, respectively. There were numerous VP1 sequences in the nucleotide database, with the lowest number of CVB1 (131 strains) and the highest number of E30 (1,763 strains). None of the serotypes had more than 200 genome sequences. The VP1 phylogenetic trees provided more complete pictures of the topology. The genome phylogenetic trees can provide clade mutations not only on VP1, but also on VP2 and VP3. Therefore, the two phylogenetic trees can verify and complement each other. In fact, the topology of the genome phylogenetic tree was almost identical to that of the VP1 phylogenetic tree, except for the lack of a few early clades due to fewer sequences. Both molecular evolution and genetic diversity analysis showed that the VP1 N-terminus, C-terminus and VP2 EF loop were the common mutation hotspot regions of all EVB, while the VP1 BC loop, C-terminus and VP3 knob region were the mutation hotspot regions of some serotypes. Interestingly, these mutation hotspot regions were distributed around the binding sites of uncoating receptors. For example, the VP1 BC loop and VP2 EF loop were key regions of receptor binding and mutation hotspots. However, the mutation sites hardly overlap with the receptor binding sites, suggesting a highly conserved receptor binding site for EVB. EVB escaped the immune barrier of the host cell through epitope mutations, but avoided effects on receptor function. This means that if the nAbs can bind to the receptor binding sites on the VP1 BC loop and VP2 EF loop, it would be difficult for the virus to escape the neutralizing effect of the antibodies by mutation. This is because mutations are likely to reduce the ability to bind to the receptor, leading to reduced viral fitness. If nAb can bind like a receptor to mutation cold spot regions (the interior and bottom of the canyon), immune escape of the mutation is harder to occur theoretically, such as the nAb 4B10 to E30 [15]. This provided important information for the development of therapeutic antibodies.

We found that the VP1 N-terminus was a high-frequency mutation region in EVB. The VP1 N-terminus was usually embedded inside the capsid, so it was not a predicted epitope obtained by our algorithm. However, some studies [52,53] have suggested that the VP1 N-terminus of enterovirus was a "breathing epitope", which will extend to the viral capsid surface and act as an epitope when the virus changes its conformation at a specific stage. In addition, after enterovirus (such as EV71 and CVA16) bound to the uncoating receptor, a series of conformational changes occurred in the viral capsid to form the uncoating mediator. The VP1 N-terminus and VP4 were the first to be transferred from the inside of the capsid to the outside via the twofold axis channel, and then the RNA genome was transported into the host cell. High-frequency mutations in VP1 N-terminus may be intended to avoid immune attack during these processes.

In summary, the VP1 BC loop, C-terminus and VP2 EF loop were the most dominant epitope and mutation hotspot regions that determine the formation of evolutionary clades of EVB. The VP1 BC loop and VP2 EF loop may be key epitope regions that determined the use

of the uncoating receptors. The study of EVB epitopes will be helpful for the monitoring and early warning of EVB epidemics and developing therapeutic antibodies.

## Supporting information

**S1 Fig. Primary and secondary structures and predicted epitopes of EVB.** Sequence alignments of VP1 (A), VP2 (B) and VP3 (C). The secondary structure elements (β-sheets and α-helices) are shown above the corresponding alignments as arrows (β-sheets) and spirals (α-helices), respectively. Predicted epitopes are shown as black circles (core epitopes) and black triangles (surrounding epitopes).
(TIF)

**S2 Fig. EVB evolutionary relationships based on viral protein sequences and 3D structures.**
(TIF)

**S3 Fig. Sequence comparison results of the VP1 DE loop and VP2 EF loop of EVB.** The secondary structure is labeled the same as in S1 Fig.
(TIF)

**S4 Fig. Antigenic similarity based on the VP1 BC loop and VP2 EF loop.**
(TIF)

**S5 Fig. Phylogenetic trees of CVB1.** CVB1 time-scale phylogenetic trees based on VP1 sequences (A) and genome sequences (B).
(TIF)

**S6 Fig. Phylogenetic trees of CVB3.** CVB3 time-scale phylogenetic trees based on VP1 sequences (A) and genome sequences (B).
(TIF)

**S7 Fig. Phylogenetic trees of E6.** E6 time-scale phylogenetic trees based on VP1 sequences (A) and genome sequences (B).
(TIF)

**S8 Fig. Phylogenetic trees of E11.** E11 time-scale phylogenetic trees based on VP1 sequences (A) and genome sequences (B).
(TIF)

**S9 Fig. CVB5 signature amino acid mutations.**
(TIF)

**S1 Table. Number of EVB sequences for evolutionary analysis.**
(DOCX)

**S2 Table. Sequence identities (%) and RMSDs (Å) of EVB viral proteins.** The upper triangle shows the sequence identity (%) of the viral proteins (VP1-VP3), and the lower triangle shows their 3D structural differences (RMSD, Å).
(DOCX)

**S3 Table. Summary of clade mutations of EVB genome phylogenetic trees.** *The same mutations indicate parallel mutations or reverse mutations in different clades.
(DOCX)

**S4 Table. Proportion of epitope and VP1 N-terminus mutations in clade mutations (%).**
(DOCX)

## Author Contributions

**Conceptualization:** Hui Wang, Changzheng Dong.

**Data curation:** Hui Wang, Yulu Fang, Yongtao Jia.

**Formal analysis:** Hui Wang, Yulu Fang.

**Funding acquisition:** Changzheng Dong.

**Methodology:** Yulu Fang, Changzheng Dong.

**Writing – original draft:** Hui Wang, Jiajie Tang, Changzheng Dong.

**Writing – review & editing:** Yulu Fang, Yongtao Jia, Changzheng Dong.

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
