## [Decision Letter · Decision Letter 0]

18 Jun 2023

PONE-D-23-14925In silico epitope prediction and evolutionary analysis reveals capsid mutation patterns for enterovirus BPLOS ONE

Dear Dr. Dong,

Thank you for submitting your manuscript to PLOS ONE. After careful consideration, we feel that it has merit but does not fully meet PLOS ONE’s publication criteria as it currently stands. Therefore, we invite you to submit a revised version of the manuscript that addresses the points raised during the review process.

We look forward to receiving your revised manuscript.

Kind regards,

Benjamin M. Liu, MBBS, PhD, D(ABMM), MB(ASCP)

Academic Editor

PLOS ONE

Journal Requirements:

Reviewers' comments:

Reviewer's Responses to Questions

**Comments to the Author**

1. Is the manuscript technically sound, and do the data support the conclusions?

Reviewer #1: Partly

Reviewer #2: Yes

Reviewer #3: Yes

2. Has the statistical analysis been performed appropriately and rigorously? 

Reviewer #1: N/A

Reviewer #2: N/A

Reviewer #3: Yes

3. Have the authors made all data underlying the findings in their manuscript fully available?

Reviewer #1: Yes

Reviewer #2: Yes

Reviewer #3: Yes

4. Is the manuscript presented in an intelligible fashion and written in standard English?

Reviewer #1: Yes

Reviewer #2: Yes

Reviewer #3: Yes

5. Review Comments to the Author

Reviewer #1: The article (PONE-D-23-14925) entitled “In silico epitope prediction and evolutionary analysis reveals capsid mutation patterns for enterovirus B” by Hui Wanget al is an interesting study that bears medical significance. My queries and suggestion regarding this manuscript are following;

INTRODUCTION:

Line 54: Write in short about the classification of enterovirus.

Line 89: Enterocirues are RNA virus. RNS viruses are susceptible to mutation, especially in their epitope region. If there are any genetic variants (genotypes) of the study types (E6, E11, E30, CVB1, CVB3, and CVB5), mention them in the introduction.

MATERIALS AND METHODS:

Line 115: Write in detail about conformational epitope prediction.

Line 129: Write in detail how the author checked the reliability of the predicted epitope (include the website)

RESULT:

Well written.

DISCUSSION:

As per the study, if the VP1 BC loop, C-terminus and VP2 EF loop are mutation hotspots. How the therapeutic antibodies will be effective against those serotypes?

Reviewer #2: 50: the phrase ''will should'' should be changed to ''may be''

102-103: should be re-cast

116: The statement should be made specific

132-133: Since the work is done by your team, try to make your statement specific

204: same as the above

444: the tense there should be changed to future tense

Reviewer #3: The authors have done good work on the title ““In silico epitope prediction and evolutionary analysis reveals capsid mutation patterns for enterovirus B”. It will add new knowledge and new areas of research to the subject area compared with other published material.

However, i have some minor concerns:

1. Moderate editing is required throughout the manuscript, for example:

1. In the introduction section, “According to the 2014-2016 surveillance report of the US Centers for Disease Control and Prevention, six of the top ten enteroviruses most frequently causing outbreaks belonged to EVB”. Moderate editing is required.

2. In the introduction section, “The uncoating receptors of Echo and CVB are Human Neonatal Fc Receptor (FcRn) and Coxsackie-adenovirus receptor (CAR), respectively both of which bind to the virus by inserting into the canyon.”. Moderate editing is required.

3. In the introduction section “The nAbs 4B10 and 6C5 bind to the interior and northern rim of the canyon of E30 respectively, both of which block the specific binding of E30 to FcRn and thus exert neutralizing activity.” Moderate editing is required.

4. In the section of Conformational epitopes prediction, “Now we use it to predict the epitopes of EVB.”. Moderate rephrasing is required.

5. In the section of Conformational epitopes prediction, “Constructing the multiple chains. Each chain consisted of VP1-VP3 and Chain 2/3/4 surrounded by the central chain Chain 1. (2) Predicting epitopes of EVB based on the multiple chains by the three algorithms (Epitopia, Ellipro, and DiscoTope) [26-28]. (3) Screening the amino acid residues on the relatively exposed surface of the viral capsid from Chain 1. Those residues, simultaneously predicted by all three algorithms, were defined as core epitopes. The surrounding epitopes were required to meet two conditions: (1) Be Predicted as candidate epitopes by any two algorithms; (2) B”. Moderate editing is required.

Best regards,

Dr. Mai Abdel Haleem Abusalah

Faculty of Medical Allied Science,

Zarqa University,

Zarqa, 13110, Jordan.

Tel: +962-796862347

e-mail: ellamomo88@yahoo.com

6. PLOS authors have the option to publish the peer review history of their article (what does this mean?). If published, this will include your full peer review and any attached files.

Reviewer #1: **Yes: **Arun Kumar Adhikary

Reviewer #2: **Yes: **Pelumi Daniel Adewole

Reviewer #3: **Yes: **Dr. Mai Abdel Haleem Abusalah

---

## [Author Response · Author response to Decision Letter 0]

31 Jul 2023

Dear reviewers and editors:

I sincerely appreciate your valuable comments and support. Your expertise and insights have contributed significantly to the improvement of my manuscript. 

This study selected six common serotypes as the representative serotypes of enterovirus B (EVB). We systematically predicted EVB conformational epitopes using the prediction algorithm previously developed in our lab, and evaluated the prediction reliability through roadmaps. The results showed that the VP1 BC loop, C-terminus and VP2 EF loop were the main regions of EVB epitopes. Further, by constructing the phylogenetic trees of EVB, it is found that epitope mutations accompanied the emergence of almost every evolutionary clade. Finally, roadmaps revealed that most of the clade mutations during EVB evolution were located on or near epitopes. However, mutations in receptor binding sites (interior and bottom of the canyon) were rare. This suggested that EVB promotes viral epidemics by breaking the immune barrier through epitope mutations, but the mutations avoided the receptor binding sites. This provided technical support for early warning of outbreaks and important information for the development of therapeutic antibodies. Neutralizing antibodies (nAbs) that bind to receptor binding sites (interior and bottom of the canyon) can slow down the effect of mutations on the neutralizing effect of antibodies, such as the nAb 4B10 to E30 (Wang et al. Nature Communications 2020).

Sincerely yours,

Changzheng Dong, Ph.D.

School of Public Health, Health Science Center, Ningbo University , P. R. China 

dongchangzheng@nbu.edu.cn

Response to reviewers

Reviewer #1:

Q1. Line 54: Write in short about the classification of enterovirus.

Authors’ response: Thanks for the suggestion. We have added the classification of enterovirus to the manuscript (Line 57-59 of Manuscript).

Q2. Line 89: Enterocirues are RNA virus. RNS viruses are susceptible to mutation, especially in their epitope region. If there are any genetic variants (genotypes) of the study types (E6, E11, E30, CVB1, CVB3, and CVB5), mention them in the introduction.

Authors’ response: Thanks for the suggestion. We have added relevant content to the manuscript (Line 97-100 of Manuscript).

Q3. Line 115: Write in detail about conformational epitope prediction.

Authors’ response: Thanks for the suggestion. We have added prediction algorithm details (Line 130-144 of Manuscript).

Q4. Line 129: Write in detail how the author checked the reliability of the predicted epitope (include the website)

Authors’ response: Thanks for the suggestion. We have revised relevant content to the manuscript. The reliability evaluation of the prediction algorithm and results mainly relies on the data of binding sites of antibodies and receptors (experimental epitopes) in the literature. The prediction reliability can be evaluated by comparing the consistency of predicted epitopes and experimental epitopes. In the previous study, by evaluating the prediction results of serotypes PV1, EVA71, CVA16 and EVD68 (Wang et al. Scientific report 2021, Rong et al. PLoS One 2021, Fang et al. Infection, Genetics and Evolution 2021), we found that the prediction algorithm and results had high reliability. In this study, in the Reliability of predicted epitopes section of Materials and methods chapter (Line 146 of Manuscript), we described the method for evaluating EVB prediction epitopes reliability. All currently reported experimental epitopes of EVB were collected from the literature and compared them with the predicted epitopes. For a more visual comparison, the 3D structure of the viral capsid in the form of a sphere was projected onto a flat surface (roadmap) using RIVEM software to compare the distribution of the footprints of predicted and experimental epitopes. A detailed comparison was made in the Predictive reliability of epitopes section of Result chapter (Line 229 of Manuscript), and it was found that the experimental epitopes overlapped highly with the predicted epitopes. Furthermore, some new prediction epitopes were identified for further experimental verification. In addition, the websites mentioned in the algorithm are described in detail in Line 134-136 of Manuscript.

Q5. RESULT:Well written.

Authors’ response: Thanks for the comments.

Q6. As per the study, if the VP1 BC loop, C-terminus and VP2 EF loop are mutation hotspots. How the therapeutic antibodies will be effective against those serotypes?

Authors’ response: This is a very critical question, and it is also the key to the development of therapeutic antibodies. Our study found that “the VP1 BC loop and VP2 EF loop were key regions of receptor binding and mutation hotspots, but the mutation sites hardly overlap with the receptor binding sites, suggesting a highly conserved receptor binding site for EVB” (Line 464-466 of Manuscript), “EVB mutation hotspot regions were distributed around the binding sites of uncoating receptors” (Line 462-463 of Manuscript). This means that if the therapeutic antibodies can bind to the receptor binding sites on the VP1 BC loop and VP2 EF loop, it would be difficult for the virus to escape the neutralizing effect of the antibodies by mutation. This is because mutations are likely to reduce the ability to bind to the receptor, leading to reduced viral fitness. The binding sites of nAbs 6C5 and 4B10 to E30 have a few residues located on the receptor binding sites. If nAb can bind like a receptor to mutation cold spot regions (the interior and bottom of the canyon), immune escape of the mutation is harder to occur theoretically, such as the nAb 4B10 to E30 (Wang et al. Nature Communications 2020). However, because antibodies are generally large, it is difficult to develop such antibodies. We have added relevant content to the manuscript (Line 466-475 of Manuscript).

Reviewer #2:

Q1. the phrase ''will should'' should be changed to ''may be''

Authors’ response: Thanks for the suggestion. We have revised it in the manuscript (Line 52 of Manuscript).

Q2. 102-103: should be re-cast

Authors’ response: Thanks for the suggestion. We have revised it in the manuscript (Line 111-112 of Manuscript).

Q3. 116:The statement should be made specific

Authors’ response: Thanks for the suggestion. We have added prediction algorithm details (Line 130-144 of Manuscript).

Q4. 132-133: Since the work is done by your team, try to make your statement specific

Authors’ response: Thanks for the suggestion. We have added relevant content to the manuscript (Line 147-157 of Manuscript).

Q5. 204: same as the above

Authors’ response: Thanks for the comments. We have revised it in the manuscript (Line 230-231 of Manuscript).

Q6. 444: the tense there should be changed to future tense

Authors’ response: Thanks for the suggestion. We have revised it in the manuscript (Line 492 of Manuscript).

Reviewer #3:

Q1. Moderate editing is required throughout the manuscript, for example:

1. In the introduction section, “According to the 2014-2016 surveillance report of the US Centers for Disease Control and Prevention, six of the top ten enteroviruses most frequently causing outbreaks belonged to EVB”. Moderate editing is required.

2.In the introduction section, “The uncoating receptors of Echo and CVB are Human Neonatal Fc Receptor (FcRn) and Coxsackie-adenovirus receptor (CAR), respectively both of which bind to the virus by inserting into the canyon.”. Moderate editing is required.

3. In the introduction section “The nAbs 4B10 and 6C5 bind to the interior and northern rim of the canyon of E30 respectively, both of which block the specific binding of E30 to FcRn and thus exert neutralizing activity.” Moderate editing is required.

4.In the section of Conformational epitopes prediction, “Now we use it to predict the epitopes of EVB.”. Moderate rephrasing is required.

5. In the section of Conformational epitopes prediction, “Constructing the multiple chains. Each chain consisted of VP1-VP3 and Chain 2/3/4 surrounded by the central chain Chain 1. (2) Predicting epitopes of EVB based on the multiple chains by the three algorithms (Epitopia, Ellipro, and DiscoTope) [26-28]. (3) Screening the amino acid residues on the relatively exposed surface of the viral capsid from Chain 1. Those residues, simultaneously predicted by all three algorithms, were defined as core epitopes. The surrounding epitopes were required to meet two conditions: (1) Be Predicted as candidate epitopes by any two algorithms; (2) B”. Moderate editing is required.

Authors’ response: Thanks for the suggestion. We have revised them in the manuscript (Line 63-66, 82-85, 88-90, 129, 130-144 of Manuscript). In addition, we have made further revisions to the full text of the manuscript.

---

## [Decision Letter · Decision Letter 1]

11 Aug 2023

In silico epitope prediction and evolutionary analysis reveals capsid mutation patterns for enterovirus B

PONE-D-23-14925R1

Dear Dr. Dong,

We’re pleased to inform you that your manuscript has been judged scientifically suitable for publication and will be formally accepted for publication once it meets all outstanding technical requirements.

Kind regards,

Benjamin M. Liu, MBBS, PhD, D(ABMM), MB(ASCP)

Academic Editor

PLOS ONE

Additional Editor Comments (optional):

Reviewers' comments:

Reviewer's Responses to Questions

**Comments to the Author**

1. If the authors have adequately addressed your comments raised in a previous round of review and you feel that this manuscript is now acceptable for publication, you may indicate that here to bypass the “Comments to the Author” section, enter your conflict of interest statement in the “Confidential to Editor” section, and submit your "Accept" recommendation.

Reviewer #1: All comments have been addressed

Reviewer #2: All comments have been addressed

Reviewer #3: All comments have been addressed

2. Is the manuscript technically sound, and do the data support the conclusions?

Reviewer #1: Yes

Reviewer #2: Yes

Reviewer #3: Yes

3. Has the statistical analysis been performed appropriately and rigorously? 

Reviewer #1: N/A

Reviewer #2: Yes

Reviewer #3: Yes

4. Have the authors made all data underlying the findings in their manuscript fully available?

Reviewer #1: Yes

Reviewer #2: Yes

Reviewer #3: Yes

5. Is the manuscript presented in an intelligible fashion and written in standard English?

Reviewer #1: Yes

Reviewer #2: Yes

Reviewer #3: Yes

6. Review Comments to the Author

Reviewer #1: I have gone through the revised manuscript (PONE-D-23-14925_R1) entitled “In silico epitope prediction and evolutionary analysis reveals capsid mutation patterns for enterovirus B”. The authors addressed my doubts satisfactorily. I have no further queries regarding the manuscript.

Reviewer #2: The manuscript is technically sound, the statistical analysis was performed appropriately and support the conclusions

Reviewer #3: The authors have done good work on the title ““In silico epitope prediction and evolutionary analysis reveals capsid mutation patterns for enterovirus B”. It will add new knowledge and new areas of research to the subject area compared with other published material.

The authors have adequately addressed all comments and performed the required amendments; hence I highly recommend accepting this interesting article.

Best regards,

Dr. Mai Abdel Haleem Abusalah

Faculty of Medical Allied Science,

Zarqa University,

Zarqa, 13110, Jordan.

Tel: +962-796862347

e-mail: ellamomo88@yahoo.com

7. PLOS authors have the option to publish the peer review history of their article (what does this mean?). If published, this will include your full peer review and any attached files.

Reviewer #1: **Yes: **Arun Kumar Adhikary

Reviewer #2: **Yes: **Pelumi Daniel Adewole

Reviewer #3: **Yes: **MAI ABDEL HALEEM ABUSALAH

---

## [Editor Report · Acceptance letter]

18 Aug 2023

PONE-D-23-14925R1 

*In silico* epitope prediction and evolutionary analysis reveals capsid mutation patterns for enterovirus B 

Dear Dr. Dong:

I'm pleased to inform you that your manuscript has been deemed suitable for publication in PLOS ONE. Congratulations! Your manuscript is now with our production department. 

Kind regards, 

on behalf of

Dr. Benjamin M. Liu 

Academic Editor

PLOS ONE